# Topologically correct synthetic reconstruction of pathogen social behavior found during *Yersinia* growth in deep tissue sites

**Stacie A Clark[1,2], Derek Thibault[3], Lauren M Shull[1,2], Kimberly M Davis[4], Emily Aunins[1,5], Tim van Opijnen[3]\*, Ralph Isberg[1,5]\***

[1]Department of Molecular Biology and Microbiology, Tufts University Graduate School of Biomedical Sciences, Boston, United States; [2]Graduate Program in Molecular Microbiology, Tufts University Graduate School of Biomedical Sciences, Boston, United States; [3]Department of Biology, Boston College, Boston, United States; [4]W. Harry Feinstone Department of Molecular Microbiology and Immunology Johns Hopkins Bloomberg School of Public Health, Baltimore, United States; [5]Tufts University School of Medicine, Boston, United States

**Abstract** Within deep tissue sites, extracellular bacterial pathogens often replicate in clusters that are surrounded by immune cells. Disease is modulated by interbacterial interactions as well as bacterial-host cell interactions resulting in microbial growth, phagocytic attack and secretion of host antimicrobial factors. To overcome the limited ability to manipulate these infection sites, we established a system for *Yersinia pseudotuberculosis* (*Yptb*) growth in microfluidics-driven microdroplets that regenerates microbial social behavior in tissues. Chemical generation of nitric oxide (NO) in the absence of immune cells was sufficient to reconstruct microbial social behavior, as witnessed by expression of the NO-inactivating protein Hmp on the extreme periphery of microcolonies, mimicking spatial regulation in tissues. Similarly, activated macrophages that expressed inducible NO synthase (iNOS) drove peripheral expression of Hmp, allowing regeneration of social behavior observed in tissues. These results argue that topologically correct microbial tissue growth and associated social behavior can be reconstructed in culture.

**\*For correspondence:**
vanopijn@bc.edu (TO);
ralph.isberg@tufts.edu (RI)

**Competing interests:** The authors declare that no competing interests exist.

## Introduction

A variety of bacterial pathogens colonize and replicate within tissues despite the presence of the host immune system (*Carter and Collins, 1974*; *Cheng et al., 2009*; *Simonet et al., 1990*). Growth in tissue sites involves the formation of distinct foci of replication, which can develop into either abscesses, granulomas, or poorly defined clusters of bacteria (*Cheng et al., 2011*; *Pagán and Ramakrishnan, 2018*). Extracellular bacterial pathogens, in particular, can establish a tissue niche and replicate to high numbers. These clusters of bacteria in tissues are often clonal, result in distinct microcolonies, and are surrounded by host innate immune cells.

*Yersinia pseudotuberculosis* (*Yptb*) is an enteric pathogen that replicates in the intestinal lumen and regional lymph nodes, with the potential for disseminating via a poorly characterized route into deep tissue sites such as the liver or spleen (*Barnes et al., 2006*). Once colonized, *Yptb* establishes extracellular foci of replication, resulting in the formation of microcolonies that develop into lesions that are densely populated by immune cells (*Simonet et al., 1990*). Within the murine spleen, the bacterium sets up a beachhead in which distinct microcolonies are derived from a single seeding bacterium (*Davis et al., 2015*). Surrounding the bacterial microcolony, which contains between 50–

5000 bacteria, are strata of immune cells. In direct contact with the population center are neutrophils, which have cytoskeletal elements that are paralyzed by the abutting bacteria that translocate Type III Secretion System (TTSS) effectors. As a consequence, phagocytosis of the pathogen by surrounding neutrophils is severely disrupted and reactive oxygen species (ROS) production is greatly reduced (*Songsungthong et al., 2010*). The resulting neutrophil–bacterium interface appears to result in a stable relationship, in which frustrated phagocytes bind, but do not internalize the nearby bacteria. Surrounding this sphere of neutrophils are macrophages, monocytes, and other immune cells, which appear to be recruited as a response to failure to clear the focus of infection.

The recruitment of neutrophils and macrophages around a cluster of *Yptb* exposes the bacterial population to different microenvironments, depending on the locale within the microcolony occupied by individual bacteria. This drives the phenotypic specialization of different microbial regulatory subpopulations that counter the host attack. Peripheral bacteria that are in direct contact with neutrophils upregulate their T3SS to facilitate disruption of neutrophil function and allow a ring of ineffective immune cells to surround the infection site (*Davis et al., 2015*). These bacteria are also part of a larger peripheral subpopulation of bacteria that neutralize nitric oxide (NO) gas being produced by the surrounding macrophages, resulting in upregulation of the NO-detoxifying protein, Hmp. The majority of the bacterial peripheral subpopulation that responds to NO is not in direct contact with any host cells, meaning that there is a specialized subpopulation that does not directly interface with host cells, but which responds to humoral anti-microbial factors. Importantly, this peripheral subpopulation participates in social behavior, as its ability to inactivate NO via the Hmp protein protects bacteria in the core of the microcolony from NO stress. In the absence of Hmp protein, the entire microcolony is exposed to NO attack, with the consequence that the microcolony eventually disintegrates and the bacteria are killed in tissues (*Davis et al., 2015*).

Although live animal models provide valuable insight into the spatial organization of bacterial subpopulations and immune cells in tissue, analysis is currently limited to microscopy of fixed tissue, with little opportunity to study the dynamics of the infection process or perform genetic analysis. Tissue culture models rely on infection of cells within a monolayer, which fail to reproduce the architecture of bacterial microcolonies surrounded by immune cells in host tissue sites. There has been limited analysis of the dynamics of bacterial growth in tissue, investigation of the spatial relationship between immune cells and the bacteria, or identification of inter-bacterial communication within the growing microbial populations in tissues. To bridge the gap between animal infection models and tissue culture models, we developed a chemical strategy to incorporate the architecture of inflammatory sites in a tissue culture system.

In this report, we develop an in vitro system that uses droplet-based microfluidics to generate matrix-embedded *Yptb* microcolonies encompassed by host innate immune cells. Social behavior of *Yptb* could be reproduced in the presence of a chemical NO generator, indicating that production of RNI by host innate immune cells does not require directional targeting of the bacteria by these cells, but rather is a consequence of multi-cell collaboration to generate toxic amounts of RNI. We observed similar results using activated macrophages that express iNOS, mimicking social behavior in tissue. The strength of this approach is that this type of topological analysis of *Yptb*-host cell interaction is unattainable in murine and tissue culture models of infection, allowing for development of high throughput screens not accessible using current culture systems.

## Results

### Droplet gels support clonal growth of *Yersinia pseudotuberculosis* microcolonies

*Yptb* microcolonies in the murine spleen are derived from single isolated bacteria that replicate as extracellular clusters and become surrounded by innate immune cells (*Davis et al., 2015*; *Simonet et al., 1990*). In order to accurately model the growth of *Yptb* in deep tissue sites, a single bacterium needs to be isolated and grown into a microcolony in a matrix that supports this 3-dimensional (3D) topology. To encase single bacterial cells in matrix, we utilized droplet-based microfluidics, which is commonly used for isolating single mammalian cells (*Köster et al., 2008*; *Macosko et al., 2015*; *Mazutis et al., 2013*).

*Yptb* was added to a matrix consisting of molten 1% ultra-low melt agarose containing 25% HyStem-C Hydrogel. HyStem-C Hydrogel has thiol-modified hyaluronan and gelatin, allowing on-demand polymerization of these two components controlled by the addition of a thiol-reactive cross-linker (*Figure 1A*). The biomatrix was included in the droplet mixture to add components that allow multivalent adhesion sites for attachment of mammalian immune cells (*Cha et al., 2017*; *Giancotti and Ruoslahti, 1999*). The mixture was then placed in a syringe pump and under constant flow injected into a fabricated two-inlet microfluidics device, with oil introduced into the other inlet, allowing encapsulation of the bacteria in a droplet matrix surrounded by oil (*Figure 1B*; Materials and methods). At a final concentration of $5 \times 10^6$ bacteria/ml, the majority of droplets contained a single bacterium, as predicted by the Poisson distribution (*Figure 1D and E*; *Mazutis et al., 2013*; *Thibault et al., 2019*). Notably, there were two groups of droplets that were collected in the output (*Figure 1C*). The large droplets contained bacteria and were approximately 65 µm in diameter (~144 pl), while the small, satellite droplets were about 8 µm in diameter (*Figure 1C and D*). After droplet generation, HyStem-C Hydrogel was crosslinked within the agarose using Extralink, a thiol-reactive crosslinker, at room temperature and oil was removed from the droplets (Materials and methods). During oil removal, the population of small droplets was lost, and only the larger droplets remained.

In tissue, *Yptb* microcolonies form clusters (*Simonet et al., 1990*). We wanted to confirm that *Yptb* grows as a cluster inside the agarose/HyStem-C Hydrogel droplets. To this end *Yptb* expressing GFP (P*tet::gfp*, expressed from pACYC184) was encapsulated in droplets, oil was removed, and the droplets were incubated at 26°C in 2xYT broth with rotation. Microcolony growth was determined by identifying a threshold that defines edges of microcolonies in the GFP channel and determining the number of pixels in the region of interest (ROI). The data were then converted to metric scale using a stage micrometer and displayed as µm². The area of the microcolony increased over time with roughly logarithmic kinetics over a 10 hr period, indicating that the agarose/HyStem-C Hydrogel droplets support efficient *Yptb* microcolony formation (*Figure 1F*). The replicating bacteria also grew in clusters, with similar appearance to *Y. enterocolitica* grown in collagen gels (*Figure 1G*; *Freund et al., 2008*). Altogether, these results show that agarose/HyStem-C Hydrogel droplets support microcolony formation of *Yptb*, mimicking growth and topological constraints observed in tissue.

## Droplet-generated *Yersinia Pseudotuberculosis* Microcolonies require Hmp to maintain growth in the presence of NO

Upon sensing of microbes, inducible NO synthase (iNOS) is synthesized by immune cells to produce gaseous NO as an important line of host defense (*Bogdan, 2001*). NO has antimicrobial properties and can react with ROS to produce additional toxic reactive nitrogen intermediates (RNI). The bacterial protein, Hmp, detoxifies NO by converting it to nitrate ($NO_3$) which is innocuous and can be used as an electron acceptor by the pathogen (*Poole et al., 1996*; *Poole and Hughes, 2000*; *Robinson and Brynildsen, 2013*). During growth in the murine spleen, bacteria located on the periphery of microcolonies express *hmp*, and loss of Hmp attenuates *Yptb* disease (*Davis et al., 2015*). To evaluate if Hmp is also required during growth in droplet microcolonies in the presence of NO, droplet microcolonies and broth cultured bacteria were comparatively evaluated for NO sensitivity. *Yptb* WT and Δ*hmp gfp+* were encapsulated in droplets, cultured in situ for 7 hr, and exposed to varying concentrations of the NO donor DETA-NONOate. Microcolony area was measured over time through a single plane. In parallel, the two strains were grown in broth culture. The growth of the WT and Δ*hmp* microcolonies was similar in the droplets in the absence of DETA-NONOate (*Figure 2A and D*). The microcolonies derived from WT were relatively resilient to NO insult, with complete blockage of growth only observed at extreme concentrations (40 mM DETA-NONOate) (*Figure 2A*). In contrast, the Δ*hmp* strain was blocked from replicating in droplets at all concentrations of DETA-NONOate tested, indicating that Hmp is essential for growth under these conditions (*Figure 2D*). Although 8 hr of 20 mM DETA-NONOate exposure showed some interference with replication in microdroplets, the WT microcolonies were clearly larger than Δ*hmp* mutant microcolonies (*Figure 2B and E*). Similar results were observed during growth in broth culture (*Figure 2C and F*). The Δ*hmp* mutant strain was sensitive to all doses of DETA-NONOate in broth culture, while growth of the WT strain was relatively unaffected at lower doses (*Figure 2C and F*). At the highest dose used (40 mM), the microdroplet-encased WT colonies showed enhanced sensitivity relative to

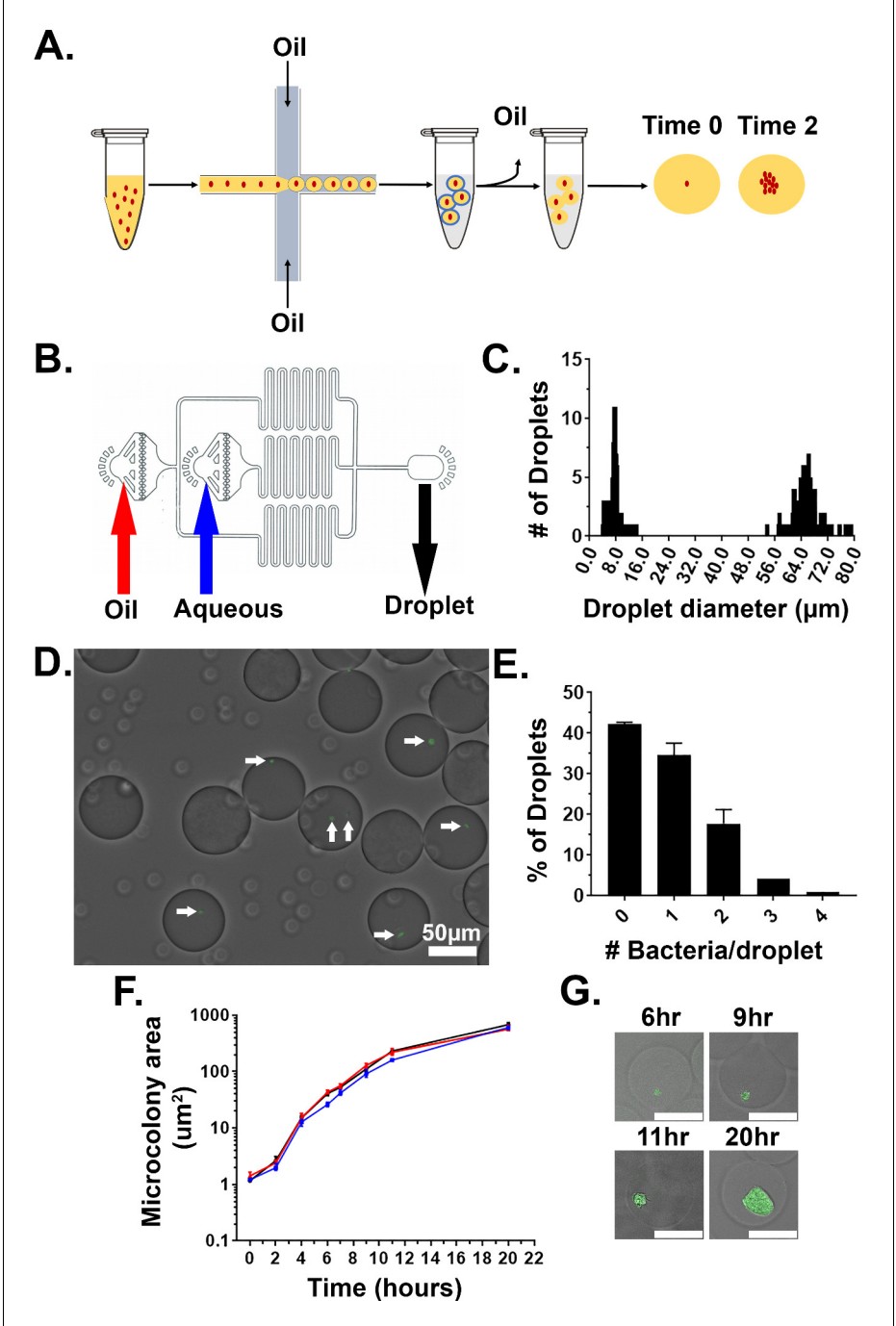

**Figure 1.** Droplet gels support clonal growth of *Yersinia pseudotuberculosis*. (**A**) *Y. pseudotuberculosis* was added to molten agarose/HyStem-C Hydrogel and introduced into a microfluidics device, allowing encapsulation within oil-coated droplets (Materials and methods). Oil was removed from the droplets and bacteria were cultured in situ within the droplets (*Thibault et al., 2019*). (**B**) Design of the microfluidics device. Noted are oil and aqueous (molten gel/bacteria mixture) phase inlets (red and blue arrows, respectively). Droplets were collected into a tube at the droplet output (black arrow) (*Thibault et al., 2019*). (**C**) Distribution of droplet sizes prior to oil removal. Droplet size was determined by capturing images from phase contrast microscopy followed by image processing (Materials and methods). (**D, E**) Droplets largely contain one or two bacterial cells immediately after encapsulation. (**D**) Image of droplets containing *Y. pseudotuberculosis gfp+*. (**E**) Presence of bacteria was scored immediately after encapsulation by phase contrast and fluorescence microscopy, scoring for GFP. (**F**) *Y. pseudotuberculosis* grows within colonies in droplets. Droplets containing encapsulated *Y. pseudotuberculosis* were cultured at 26°C, and microcolonies were visualized at indicated timepoints by phase contrast and fluorescence microscopy.

*Figure 1 continued on next page*

*Figure 1 continued*

Microcolony areas were determined by image analysis (Materials and methods). Each timepoint is median + / - 95% confidence interval (CI) of 3 biological replicates of 50 microcolonies. (**G**) Representative images of microcolonies from (**F**) at the noted times visualized by phase contrast and fluorescence microscopy. Scale bar: 50 μm.

bacteria grown in broth (*Figure 2A and C*). Nevertheless, these results indicate that Hmp is required to maintain optimum growth of microcolonies in the presence of NO.

## Topological constraints drive non-uniform expression of *hmp*

During infection by *Yptb*, subpopulations of bacteria within microcolonies respond to distinct micro-environments (*Davis et al., 2015*). Peripheral bacteria upregulate Hmp to detoxify products of iNOS generated by immune cells, while interior bacteria are protected from RNI and show no such upregulation. The droplet microcolonies are morphologically similar to microcolonies during infection, so we predicted that *hmp* would also be expressed peripherally in response to DETA-NONOate. To probe for non-uniform expression in response to DETA-NONOate, we compared the fluorescence of WT and Δ*hmp gfp+* bacteria harboring a plasmid with the promoter of *hmp* driving expression of mCherry (*Phmp::mcherry*) after exposure to DETA-NONOate using flow cytometry. *Phmp::mcherry* expression should be uniform in broth cultures of WT *Yptb*, while *hmp* should exhibit nonuniform expression in droplets if microcolony structure drives spatial regulation of *hmp* expression (*Figure 3A*: left).

To compare *Yptb* grown in droplet microcolonies to bacteria in broth cultures, a protocol was developed to disperse droplet microcolonies into single cells (*Figure 3A*; Materials and methods), to allow flow cytometry analysis of the constitutive *gfp* and *Phmp::mcherry* reporters. The WT and Δ*hmp* strains harboring *gfp⁺ Phmp::mcherry* were cultured in either droplet microcolonies or broth and exposed to 0,1, 2.5, 5, and 10 mM of DETA-NONOate for 0, 4 and 8 hr (*Figure 3B and C*, *Figure 3—figure supplement 1*) followed by flow cytometry analysis. In broth grown cultures, WT and Δ*hmp* strains harboring *Phmp::mcherry* showed fluorescence intensity increases after NONOate exposure, with a single population of both *gfp* and *Phmp* high expressers and small populations of nonexpressers in all cases (*Figure 3B*: dark-lined, open histograms). In contrast, starting at 4 hr post-exposure of DETA-NONOate to WT microcolonies growing in droplets, there was a broad distribution of *Phmp::mcherry* fluorescence compared to either broth culture or to the *gfp* fluorescence

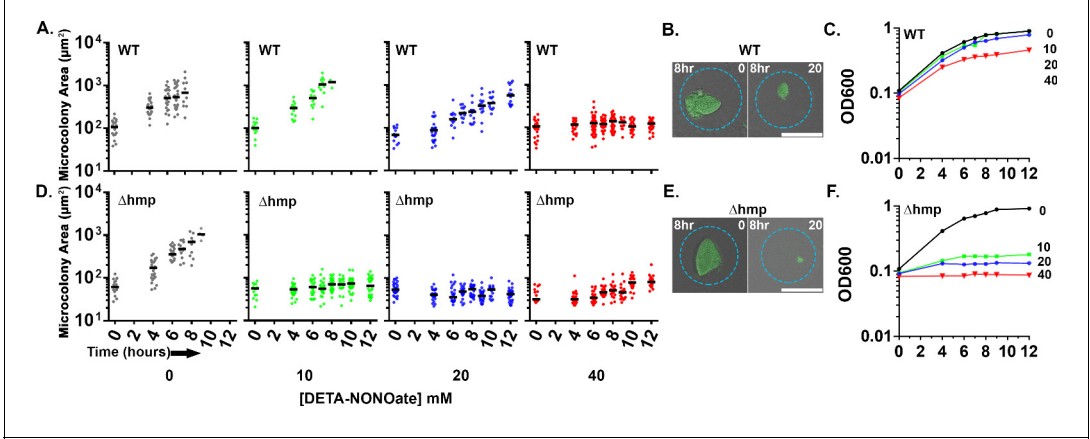

**Figure 2.** Droplet-generated *Yersinia pseudotuberculosis* microcolonies require Hmp to maintain growth in presence of NO. Indicated *Y. pseudotuberculosis* strains were cultured in 2xYT with rotation at 26˚C for 7 hr prior to DETA-NONOate exposure. At the indicated timepoints, an aliquot of droplets was fixed and visualized by fluorescence microscopy. (A, D). Microcolony area determined from captured images followed by image analysis (Materials and methods). Each point represents a single droplet with median noted. (B, E). Representative WT (B) and Δ*hmp* (E) microcolonies after 8 hr of treatment with and without DETA-NONOate. Scale bar: 50 μm. (C, F). Overnight broth cultures of *Y. pseudotuberculosis* were diluted 1:10 in 2xYT broth prior to addition of DETA-NONOate. At indicated timepoints, culture density was determined (Materials and methods).

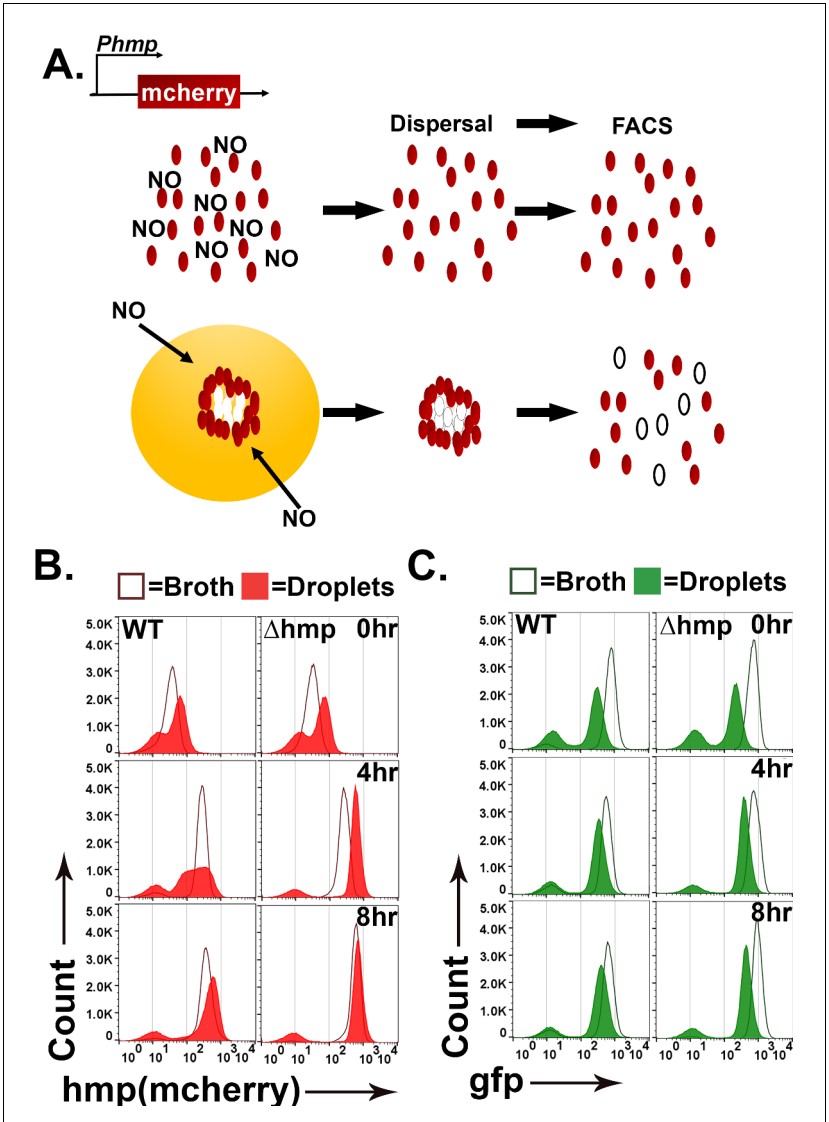

**Figure 3.** Topological constraints drive non-uniform expression of *hmp*. (A). Model of the predicted outcome for the nitric oxide (NO) response of *Y. pseudotuberculosis* grown in broth versus droplet culture (***Davis et al., 2015***). Red: bacteria showing high expression of $P_{hmp}$-*mCherry* reporter. White: bacteria showing undetectable mCherry fluorescence. (B, C). *Y. pseudotuberculosis* grown in droplet microcolonies show altered response to NO. Noted bacterial strains harboring $gfp^+P_{hmp}$-*mcherry* plasmids were grown in either broth (open histograms) or droplets (filled histograms) in the presence of NO donor for the noted times. Droplet microcolonies were dispersed at noted times and analyzed by flow cytometry (Materials and methods). Histograms represent total ungated events. B. Flow analysis of mCherry channel. C. Analysis of GFP channel.

The online version of this article includes the following figure supplement(s) for figure 3:

**Figure supplement 1.** Topological constraints drive non-uniform expression of *hmp*.

(***Figure 3B*** red-shaded profile: 4 hr and ***Figure 3—figure supplement 1A***). Continued exposure to NONOate for 8 hr partially dissipated this broad peak, as the distribution of mCherry became more uniform and intense, although the fluorescence profile of the WT strain in microcolonies was clearly broader than when grown in broth culture (***Figure 3B***: 8 hr and ***Figure 3—figure supplement 1A***). These results are consistent with microcolony structure driving heterogeneous expression of *Phmp:: cherry* in response to an RNI generator, with prolonged exposure decreasing heterogeneous firing of the promoter.

The heterogeneous *mcherry* reporter fluorescence in WT was dependent on the *hmp* promoter, as GFP expression driven by the unrepressed Tet promoter was uniform in bacteria that originate from broth culture and droplet microcolonies (*Figure 3C* and *Figure 3—figure supplement 1B*). Similar to the observations with *mcherry* there was a small population representing approximately 5% of the bacteria that failed to fire *gfp* perhaps due to the dispersal and fixation procedure. These results argue that microcolony topology is a critical factor that drives heterogeneity observed in bacterial subpopulations in response to different microenvironments.

The heterogenous distribution of *Phmp::mcherry* fluorescence was dependent on the presence of the Hmp protein. The introduction of the Δ*hmp* mutation resulted in a distribution of *hmp::mcherry* fluorescence intensity from droplet microcolonies that was similar to that observed in broth cultures, although there was more rapid firing of the reporter than in broth, with maximal fluorescence already established after 4 hr exposure to DETA-NONOate (*Figure 3B*: right and *Figure 3—figure supplement 1C*). Therefore, the entire microcolony has the potential to respond to RNI in the absence of Hmp, consistent with the Hmp protein acting to lower the amount of inducing RNI in a subpopulation of bacteria.

## Spatial regulation of *hmp* expression in tissues can be reproduced in droplet culture

We hypothesized that the heterogenous distribution of *Phmp::mcherry* fluorescence of WT *Yptb* as seen by flow cytometry analysis is indicative of spatial regulation of the promoter. To determine if the spatial regulation of *Phmp* observed in tissue can be reproduced in droplet culture, WT and Δ*hmp* strains carrying the *gfp*+ and *Phmp::mcherry* reporters were encapsulated in droplets, grown into microcolonies, and exposed to DETA-NONOate for 4 hr (*Figure 4A*). WT microcolonies showed peripheral expression of *hmp*, while microcolonies lacking Hmp show uniform expression of the *hmp* reporter (*Figure 4B and C*, respectively). Peripheral expression was dependent upon the NO-driven signal, as the constitutive GFP signal was uniform throughout the microcolonies (*Figure 4B*).

To quantitatively analyze spatial regulation of *hmp*, an automated script was developed from images captured using a 63X lens (*Figure 5A*). The script defines the periphery of microcolony and sets a mask to determine the region of interest (ROI), allowing a user-defined width extending into the microcolony (12 pixel-radius structuring elements used throughout study) and a point of lowest intensity (PLI, five pixel-radius structuring element used here) identified in each ROI (determinants displayed in *Figure 5B*). The fluorescence at the periphery is calculated as a ratio of the average pixel intensity of the mCherry channel divided by the average pixel intensity of the GFP channel in the periphery. The fluorescence at the PLI is also calculated this way and the periphery versus PLI ratio is determined. A ratio greater than one indicates preferential peripheral expression of *Phmp::mcherry*.

There was a significant increase in the periphery vs. PLI ratio in WT microcolonies in response to DETA-NONOate after 4 hr exposure at concentrations of 1 or 2.5 mM, indicating peripheral expression of *hmp::mcherry* (*Figure 6A*,left; p<0.0001). Peripheral expression in the presence vs. absence of RNI stress was independent of colony size, as there was no difference in microcolony area after DETA-NONOate exposure in WT microcolonies (*Figure 6B*: left). Microcolonies lacking Hmp, in contrast, had uniform expression of the *hmp* reporter indicated by a ratio ~1 (*Figure 6A*: right). Interestingly, the periphery vs PLI ratio in microcolonies lacking Hmp significantly decreased in the presence of DETA-NONOate (p<0.0001). This indicates that RNI likely accumulate in the center of microcolonies in the absence of Hmp, a phenomenon also observed in tissue (*Davis et al., 2015*). Microcolonies lacking Hmp were significantly smaller after DETA-NONOate exposure, further showing detoxification by Hmp is required for the growth of a microcolony (*Figure 6B*: right, p<0.0001).

Flow cytometry analysis of WT *Yptb* originating from droplet microcolonies exposed to these low DETA-NONOate levels revealed heterogenous expression when compared to bacteria that were not exposed to DETA-NONOate (*Figure 6C*: left). This argues that the broad distribution of *mcherry* fluorescence intensity is a consequence of spatially regulated *Phmp* firing (*Figure 6A*). In contrast, *Yptb* Δ*hmp* microcolonies challenged with DETA-NONOate for 4 hr revealed a tight distribution of *mcherry* fluorescence, indicating penetration of RNI through the entire microcolony (*Figure 6C*: right). Therefore, the contour of fluorescence expression from a reporter construct was diagnostic of heterogeneous expression patterns.

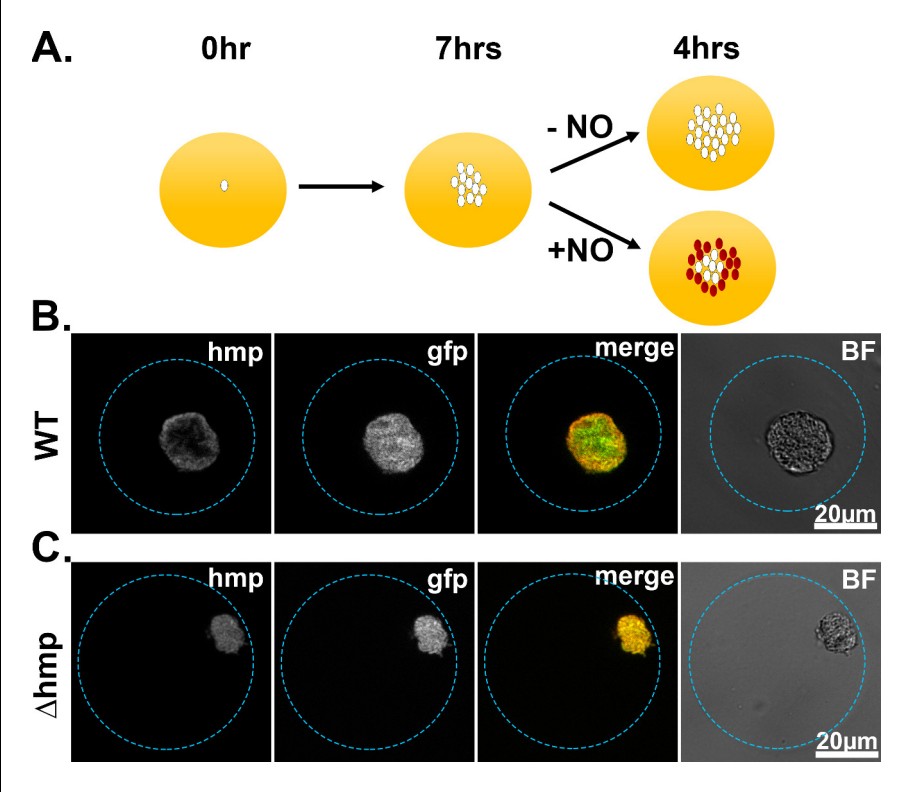

**Figure 4.** Spatial regulation of *hmp* expression in tissues can be reproduced in droplet culture. (**A**). Experimental overview. Droplets seeded with *Y. pseudotuberculosis* harboring fluorescence reporters were incubated for 7 hr at 26°C and varying concentrations of DETA-NONOate were added to the cultures for 4 hr at 37°C, 5% $CO_2$. After 4 hr, the droplets were fixed and visualized using brightfield and fluorescence microscopy. (**B, C**). Representative WT and Δ*hmp* microcolonies after 4 hr of treatment with 1 mM DETA-NONOate.

## Activated BMDMs drive peripheral expression of *hmp* in droplet microcolonies

NO gas is produced by iNOS[+] cells around the periphery of a neutrophil encased *Yptb* microcolony (*Figure 7A*). The iNOS[+] cells recruited to the inflammatory site consist of monocytes and macrophages (CD68[+]) cells and are physically separated from the bacterial microcolony (*Figure 7A*: right) (*Davis et al., 2015*). When seeding is initiated, or microcolonies are small, there are very few iNOS[+] cells near the site of bacterial replication (*Figure 7A*; Seeding Event, Neutrophil Recruitment). Once the colony is completely surrounded by neutrophils, iNOS[+] cells are apparent, forming a sphere about recruited neutrophils. Recruitment of iNOS[+] cells to a site of bacterial replication as a second step in bacterial restriction is consistent with work done in other pathogens (*MacMicking et al., 1997*; *Murray and Nathan, 1999*; *Shiloh et al., 1999*; *Shiloh and Nathan, 2000*; *Vazquez-Torres et al., 2000*). The 65 μm droplets, with bone marrow-derived macrophages (BMDMs) associated with their surface, are predicted to allow extremely good mimicking of action-at-a-distance by iNOS[+] cells, as the microcolonies in the engineered system are situated at similar distances relative to the droplet surface (*Figure 7A*, iNOS[+] Cell Recruitment). To recapitulate tissue dynamics, therefore, BMDMs were activated to produce iNOS[+] and added to droplets containing pregrown microcolonies (*Figure 7B*). The agarose/HyStem-C Hydrogel droplet served as the buffer zone between the bacterial microcolony and the iNOS[+] cell layer (*Figure 7B*). Pregrowth of the microcolony prior to exposure to BMDMs is envisioned to recapitulate the kinetics observed from histology of spleen tissues (*Figure 7A*).

LPS and IFNγ were incubated with BMDM to activate iNOS, prior to adding the cells for 4 hr to droplets harboring microcolonies previously grown for 7 hr (*Figure 8A and B*; *Frawley et al., 2018*; *Green et al., 1994*; *Mosser and Gonçalves, 2015*). Efficient expression of iNOS was dependent on

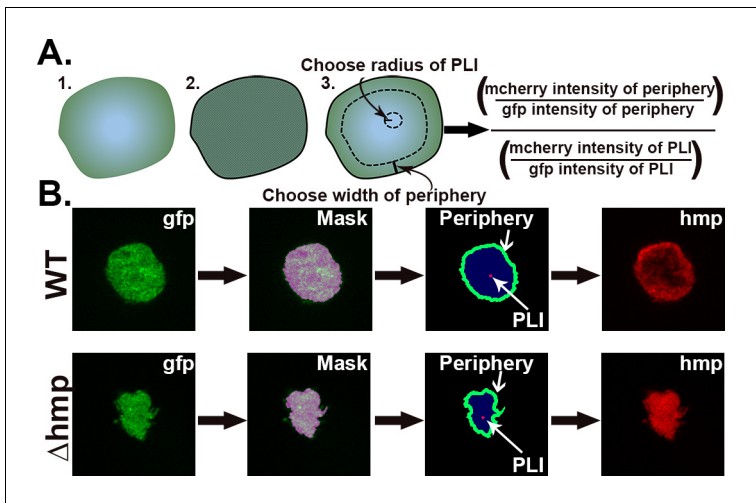

**Figure 5.** Strategy for quantification of *hmp* expression localized on periphery of microcolonies. (**A**) Design of script that allows the identification of regions of interest (ROIs) for the purpose of determining the average fluorescence intensities and normalization strategies within the ROIs. Script defines periphery of microcolony and sets mask, allowing a user-defined width extending into the microcolony along with a point of lowest intensity (PLI) which is determined based on the mCherry signal for each microcolony. The fluorescence at the periphery is calculated as a ratio of the average pixel intensity of the mcherry channel in periphery divided by the average pixel intensity of the GFP channel in the periphery, allowing normalization for bacterial expression. The fluorescence for the defined PLI is also calculated in the same fashion, and the periphery verses PLI ratio is determined (Supplemental Data). (**B**) Example of identification of periphery, mask generation and definition of peripheral ROI. The GFP channel defines the periphery, allowing mask to be set. Within the mask, the periphery is defined as extending 12 pixels into the microcolony. The PLI within the mask is defined as the area of 5-pixel radius with the lowest intensity in the mCherry channel.

LPS/IFNγ treatment, based on immunofluorescence probing with anti-iNOS (*Figure 8B*, left panel). Using this system, WT microcolonies encapsulated in droplets incubated with activated BMDMs exhibited spatially regulated expression of the *Phmp::mcherry* reporter that appeared to exactly mimic the behavior of bacteria growing in tissue (compare *Figure 8C and D*). As observed in tissues (*Davis et al., 2015*), this spatial regulation was lost in the Δ*hmp* strain in response to the addition of the activated BMDMs (*Figure 8E*). Consistent with the activated BMDMs producing anti-microbial levels of iNOS, growth of the Δ*hmp* strain was severely inhibited by the addition of BMDMs at a distance, dependent on the addition of LPS/IFNγ and the presence of active iNOS (*Figure 8F*; p<0.0001). Surprisingly, WT microcolonies were significantly larger in the presence of activated BMDMs, partially dependent on the presence of iNOS (*Figure 8F*). This is consistent with activated macrophages providing soluble growth-promoting substances at a distance, including either $NO_3$ or oxidized cofactors as a consequence of Hmp activity.

To show that peripheral expression of *hmp* was dependent upon RNI, the NOS inhibitor, L-NMMA, was introduced into this system (*Figure 8F,G*). BMDMs were incubated in the presence or absence of 2 mM L-NMMA prior to challenging droplet microcolonies, and expression of the *Phmp::mcherry* reporter was analyzed by microscopy (*Frawley et al., 2018*). In response to LPS/IFNγ-primed BMDMs, there was a large increase in peripheral expression of the *Phmp::mcherry* reporter relative to the addition of BMDMs in the absence of activation (*Figure 8G*; p<0.0001). Peripheral expression of the mCherry reporter was lost in WT microcolonies when iNOS was inhibited by NMMA, with the resulting distribution being indistinguishable from BMDMs in the absence of activation (*Figure 8G*). As expected, there was no spatial regulation of *Phmp::mcherry* in the strain lacking *hmp* activity, and the presence of NMMA had no effect (*Figure 8G*). These results indicate that peripheral expression of *Phmp::mcherry* is dependent on the presence of intact Hmp protein and in response to RNI generated by iNOS activity.

To demonstrate that there was heterogeneous *hmp* expression in WT microcolonies on a population level, WT and Δ*hmp* microcolonies challenged with unprimed, LPS/IFNγ-primed, or L-NMMA

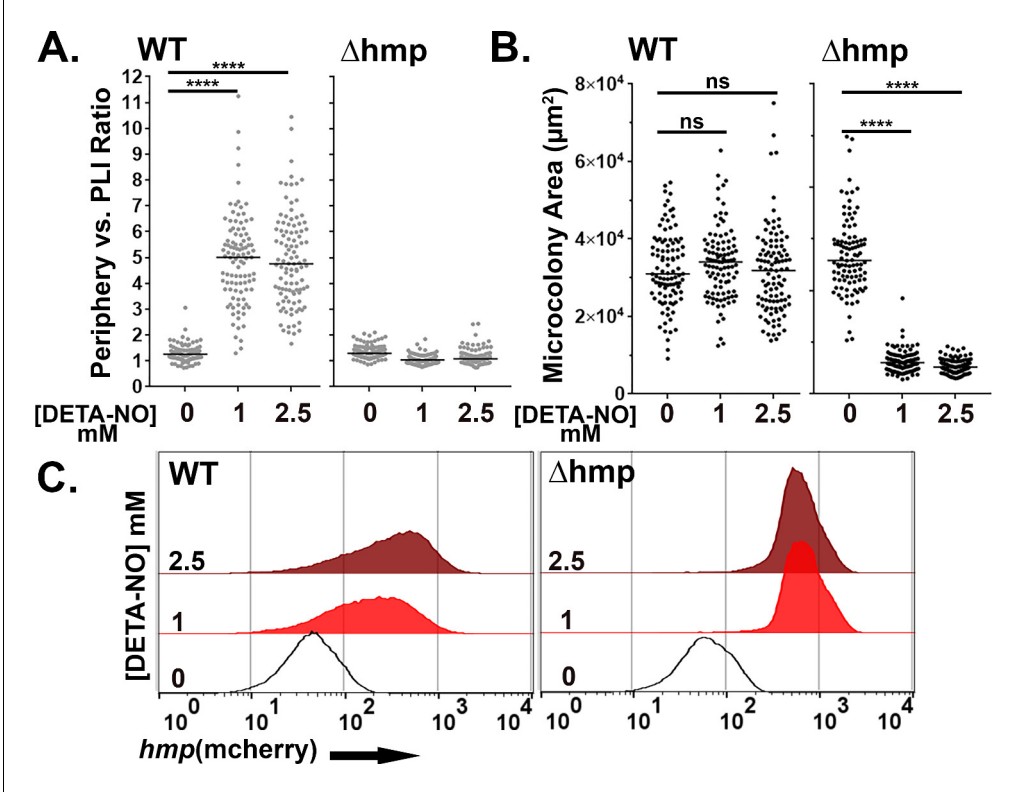

**Figure 6.** Hmp activity is required for peripheral expression of *hmp* in response to NO. (**A–C**). The presence of Hmp activity is required for *Y. pseudotuberculosis* to have enhanced expression of $P_{hmp}$ in microcolonies exposed to NO. (**A**) Periphery versus PLI ratio for microcolonies as a function of DETA-NONOate concentration in microcolonies from noted bacterial strains. (**B**) Microcolony area (pixels$^2$) for WT and Δ*hmp* microcolonies at different concentrations of DETA-NONOate. Black and gray circles: 100 individual microcolonies measured for each condition. Lines indicate the medians. (**C**). Flow cytometric analysis of dispersed microcolonies. Droplets were dispersed into single cells after 4 hr of DETA-NONOate treatment at noted concentrations and subjected to flow cytometry (Materials and methods). Bacteria were gated for GFP$^+$ and analyzed for mCherry signal (*Figure 6—figure supplement 1*). Statistics: Mann-Whitney test, ****p<0.0001, ns: not significant.

The online version of this article includes the following figure supplement(s) for figure 6:

**Figure supplement 1.** Gating strategy for flow cytometry.

---

treated-LPS/IFNγ-primed BMDMs were dispersed and analyzed by FACS (*Figure 8H*). In bacteria originating from WT microcolonies incubated with unprimed BMDMs, there were low levels of fluorescence from the *Phmp::mcherry* reporter (*Figure 8H*: left, no treatment). After challenge with LPS/IFNγ-primed BMDMs, there was increased *hmp* expression, with a small percentage of the population expressing high levels of the *Phmp::mcherry* reporter (*Figure 8H* left, LPS/IFNγ). Inhibition of activated BMDMs with NMMA resulted in a peak that was almost indistinguishable from the bacteria exposed to unprimed BMDMs (*Figure 8H* left, L-NMMA + LPS/IFNγ). Bacteria that originated from Δ*hmp* microcolonies showed uniformly high fluorescence contours for *Phmp::mcherry* after challenge with LPS/IFNγ-primed BMDMs, showing that NO secreted by macrophages can penetrate the entire microcolony (*Figure 8H*: right). As expected, there was low *Phmp::mcherry* fluorescence after challenge with unprimed BMDMs and NOS-inhibited BMDMs (*Figure 8H* right; L-NMMA + LPS/IFNγ). In summary, peripheral expression of *hmp* in WT droplet microcolonies was dependent upon challenge with LPS/IFNγ-primed BMDMs and mimicked the bacterial response to recruited innate immune cells observed in mouse tissue.

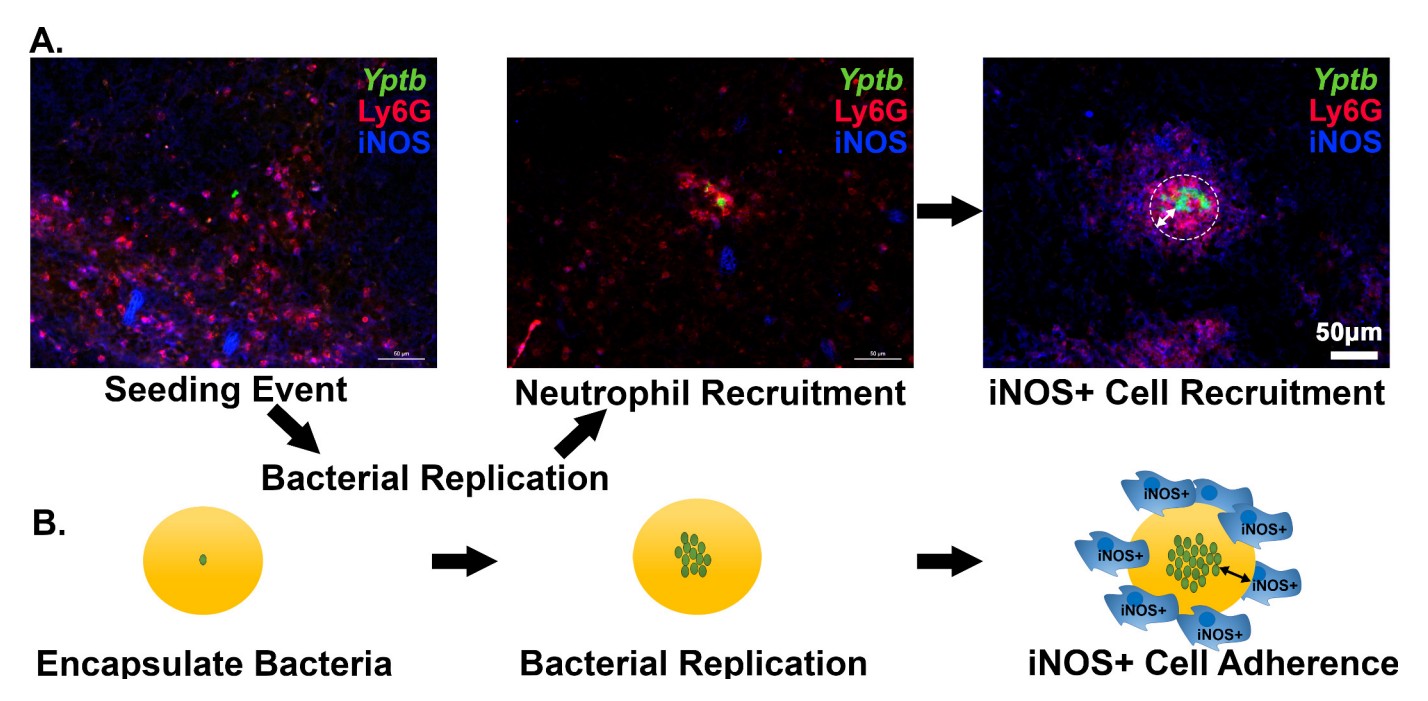

**Figure 7.** Recruitment of iNOS[+] cells is associated with the maturation of *Yersinia pseudotuberculosis* microcolonies in tissue. (**A**) C57BL/6 mice were inoculated intravenously (i.v.) with $10^3$ WT *gfp+* bacteria, and spleens were harvested 3 days post-inoculation (PI). Spleen tissue was probed with α-Ly6G and α-iNOS and visualized by fluorescence microscopy. Neutrophils (α-Ly6G[+]; red) surround a cluster of *Y. pseudotuberculosis* (green). Surrounding the sphere of bacteria that is encased by neutrophils (white circle) is a layer of iNOS+ cells. White arrows: The distance between *Y. pseudotuberculosis* and the iNOS+ cell layer (not to scale). (**B**) Model for droplet microcolonies. Bacteria are grown in situ within the droplets prior to challenge with iNOS[+] cells to mimic *Y. pseudotuberculosis* microcolony structure in tissue. Black arrows note there is space between iNOS+ cells and cluster of bacteria.

## Discussion

Many invasive bacterial pathogens can colonize hosts and disseminate to both peripheral and deep tissue sites where they establish replicative niches. For pathogens as diverse as *Mycobacterium tuberculosis* and *Staphylococcus aureus*, growth in tissues is followed by recruitment of neutrophils, macrophages and inflammatory monocytes, which can lead to the formation of abscesses or granulomas when the infection is not eradicated (*Cheng et al., 2011*; *Pagán and Ramakrishnan, 2018*). These pathogen-immune cell aggregates are intricate structures in which the pathogen interfaces with host cells, but also result in significant self and social interactions, making it difficult to reproduce in culture the entire set of events present in tissue. Although there is considerable research on infectious lesions caused by these pathogens, there are few informative culture models that reconstruct the pathogen-immune cell architecture occurring in tissues that preserve inter-microbial interactions (*Elkington et al., 2019*; *Fitzgerald et al., 2014*; *Fonseca et al., 2017*; *Guggenberger et al., 2012*; *Guirado et al., 2015*).

We have addressed this gap in the field by reconstructing *Yptb* inflammatory sites using microfluidic technology to generate a 3D model that allows interbacterial interactions to be studied as well as action-at-a-distance by immune cells, with topology mimicking a tissue infection. *Yptb* is an intestinal pathogen that can establish extracellular foci in deep tissue sites after translocation across the intestine into either the bloodstream or regional lymph nodes (*Barnes et al., 2006*). In the course of establishing an infectious niche within the murine spleen, *Yptb* forms microcolonies tightly associated with recruitment of neutrophils that directly contact the cluster of bacteria, which are, in turn, encased by layers of more neutrophils followed by macrophages and monocytes that largely do not contact bacteria directly (*Davis et al., 2015*). The model system developed here closely mimics this topology. Droplets support the growth of *Yptb* microcolonies, and activated BMDMs localized at a

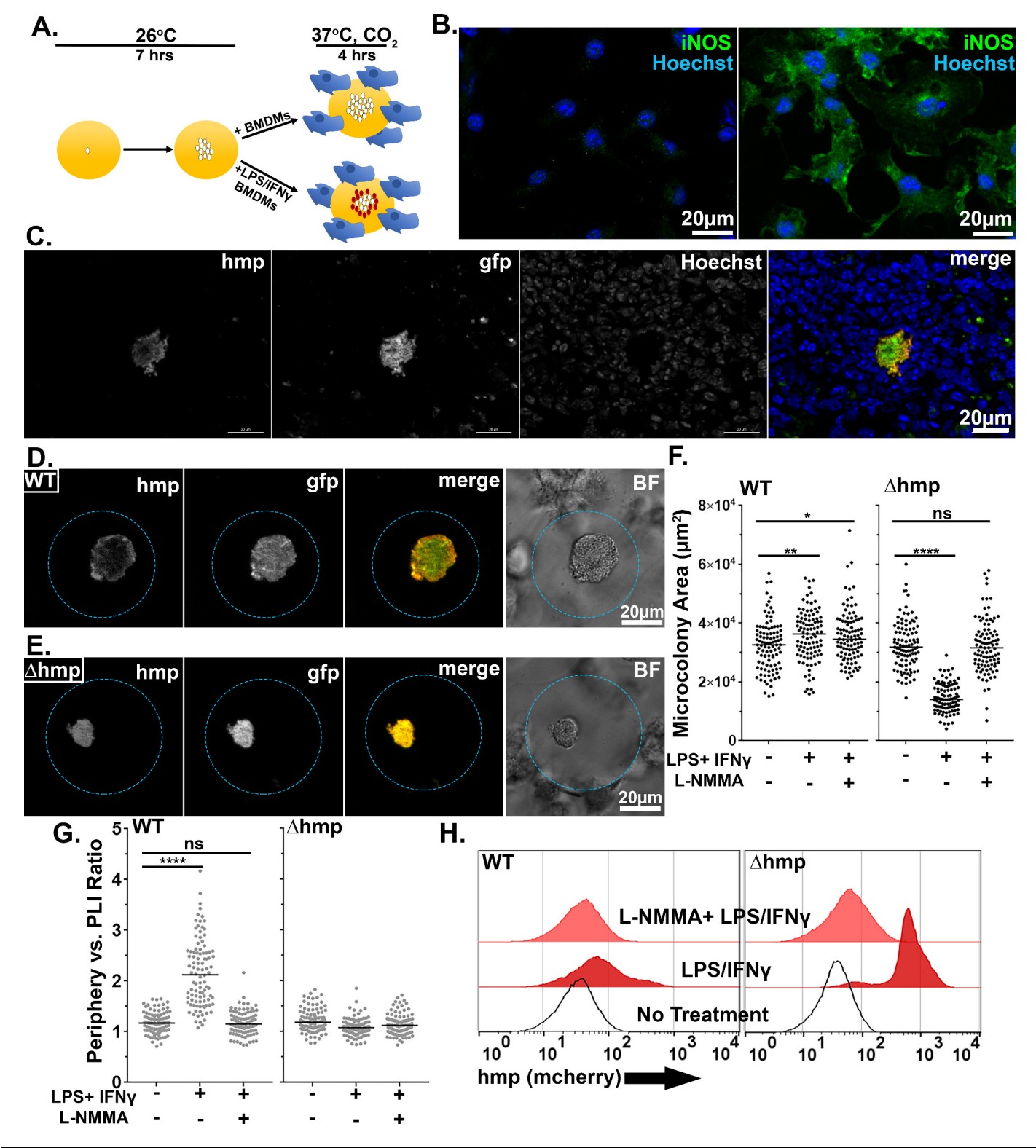

**Figure 8.** Activated BMDMs drive peripheral expression of *hmp* in droplet microcolonies, mimicking tissue spatial regulation. (**A**) Experimental overview. Droplets containing *Y. pseudotuberculosis* were grown for 7 hr at 26°C prior to incubation with BMDMs for 4 hr at 37°C, CO₂. The fixed microcolonies were visualized using brightfield and fluorescence microscopy and subjected to image analysis (Materials and methods). (**B**) Resting BMDMs (left) and LPS/IFNγ-primed BMDMs (right) were fixed, probed with α-iNOS and visualized by fluorescence microscopy. (**C**). C57BL/6 mice were inoculated intravenously (i.v.) with 10³ WT *gfp+ P_hmp-mcherry* bacteria, and spleens were harvested 3 days post-inoculation (PI) for fluorescence

*Figure 8 continued on next page*

*Figure 8 continued*

microscopy. A representative microcolony from tissue is shown. (D, E). Representative WT (D) and Δ*hmp* (E) microcolonies incubated with LPS/IFNγ-primed BMDMs. (F). Microcolony area in presence of BMDMs, incubated as noted. (G) Periphery verses PLI ratios for microcolonies incubated with BMDMs treated as noted. Black and gray circles: 100 individual microcolonies from each condition. Line indicates median. (H). Flow cytometry of dispersed microcolonies after 4 hr of incubation with BMDMs. Dispersed bacteria were gated for GFP$^+$ and mCherry fluorescence was determined (Materials and methods). Statistics: Mann-Whitney test, ****p<0.0001, ***p<0.0005, **p<0.005, *p<0.05, ns: not significant.

similar distance to that observed in tissues drive a bacterial transcriptional response that recapitulates interactions observed in tissue.

Associated with the recruitment of immune cells to infection is the presence of iNOS$^+$ cells that are clearly activated to express iNOS and release NO. These cells do not directly contact the growing colony, so their ability to modulate microbial growth would appear to be linked to the production of soluble anti-microbial mediators. In concert with attack-from-a-distance, there is an outer shell of bacteria protecting the microcolony from RNI attack by producing the detoxifying Hmp protein that clearly responds to this stress without directly contacting iNOS$^+$ cells. To mimic clonal growth in the spleen and action at distance, our tissue culture system used microfluidics-generated droplets, which allowed us to maximize the capture of single bacteria within 65 μm agarose/HyStem-C Hydrogel droplets and allow adhesion of macrophages on the droplet surface. The bacteria appeared to tolerate being encompassed in this environment as evident by *Yptb* replication, allowing a reasonable facsimile of spleen-resident microcolonies. In addition to the cells analyzed here, neutrophils are present surrounding the cluster of *Yptb*, but only a few peripheral bacteria are in direct contact with these cells, with the consequence that they are largely inhibited by translocated TTSS effector proteins. Future work will involve modifying the system to allow the incorporation of neutrophils, so that the response of bacteria on the extreme periphery of the colony can be analyzed during this process. This will involve either embedding cells in the droplet matrix, or forcing controlled degradation of the matrix to allow close contact with the microcolony (*Guggenberger et al., 2012*; *Sakai et al., 2012*).

One of the surprising results from this study was that we could totally reproduce topologically driven regulation of the *hmp* promoter by incubating droplet-encased microcolonies in the presence of the NO generator DETA-NONOate. This indicates that production of RNI by iNOS$^+$ cells does not require directional targeting of the bacteria by these cells, but rather is a consequence of multi-cell collaboration to generate toxic amounts of RNI, with the concentrations determined by the total number of recruited iNOS$^+$ cells. By encompassing the infection site, the sum of the RNI produced by individual cells generates a sufficient amount of chemical to induce transcription of the reporter construct and kill bacterial mutants unable to produce Hmp.

To demonstrate that the addition of this chemical could reconstruct RNI signaling within the bacterial colony, we developed a script that could be used to quantify the accumulation of a fluorescence signal around the periphery of microcolonies relative to the constitutive GFP control. This strategy demonstrated that the use of DETA-NONOate was sufficient to generate a robust response on the periphery relative to a region internal to the colony that defined the area of lowest fluorescence intensity. While variation in copy number of the reporter plasmids could influence heterogeneity of fluorescence induction, previous work has shown that the *hmp* mutant strain rescued with *hmp-mcherry* integrated on the chromosome also results in spatial regulation of *mcherry* within the murine spleen indicating that copy number does not seem to play a role (*Davis et al., 2015*). It also demonstrated that the gradient of *Phmp* expression depends on both iNOS$^+$ activity and overall activation of macrophages by LPS and IFNγ. Based on quantification of this ratio, we would argue that addition of BMDMs to the droplets generated somewhat lower amount of RNI than resulted from 1 mM DETA-NONOate (compare *Figures 6A–8G*). It would be difficult to directly compare these results to mouse tissue, as the nature of the samples and preservation technique differs greatly from the droplet analysis. It does argue that the amount of RNI generated by 1 mM DETA-NONOate is more than sufficient to explain the images generated from the mouse spleen.

One of the strengths of the synthetic droplet system is that we were able to disrupt the droplets to free the bacteria and analyze bacterial populations in bulk, a task that is often difficult to perform when isolating colonies from mammalian tissues. Once the colonies were disrupted, we demonstrated that individual cells can be subjected to flow cytometry analysis, allowing the response of

spatially distinct population of bacteria to be analyzed. This approach should facilitate identifying additional macrophage secreted products that can penetrate microcolonies and control bacterial gene expression. There are a variety of small secretory molecule products of macrophages that have been identified, but the roles of these products in controlling bacterial growth or modulating gene expression of microorganisms is greatly understudied (*Nathan, 1987*; *Sugimoto et al., 2012*). Presumably, transcriptional analysis of bacterial subpopulations will facilitate dissection of previously unappreciated bacterial responses to the host and potentially identify new host strategies for restricting microbial growth.

## Materials and methods

### Bacterial strains and growth conditions

All experiments used derivatives of the *Y. pseudotuberculosis* strain IP2666 (*Balada-Llasat and Mecsas, 2006*). For all droplet experiments, bacteria were grown overnight into stationary phase in 2xYT broth (*Crimmins et al., 2012*; *Davis et al., 2015*) at 26˚C with rotation. Overnight cultures were diluted 1:2000 in 1% ultra-low-melt agarose (Sigma, USA, #A2576) in 2xYT containing 25% Hystem-C hydrogel kit (ESI BIO, CA, USA, #GS312) prior to droplet generation. For broth NO exposure, overnight cultures were diluted 1:100 in 2xYT and rotated at 26˚C for the indicated times.

### Generation of plasmid-based reporter strains

The *Y. pseudotuberculosis* reporter strains in this study have been previously described (*Crimmins et al., 2012*; *Davis et al., 2015*). GFP$^+$ strains express GFP from the unrepressed *PTet* promoter of pACYC184 (*Chang and Cohen, 1978*). The *Phmp::mCherry* transcriptional fusion was constructed by fusing the *hmp* promoter to *mCherry* using overlap extension PCR, and cloning into the pMMB67EH plasmid (*Fürste et al., 1986*). Strains that express both reporters are co-transformants of the compatible pACYC184 and pMMB67EH plasmids.

### Droplet generation

CAD-designed microfluidics chips having 38 independent devices controlled by two input ports were constructed according to published protocols (*Mazutis et al., 2013*; *Thibault et al., 2019*) at the Boston College Nanofabrication Facility (https://www.bc.edu/bc-web/schools/mcas/sites/cleanroom.html). After construction, each microfluidics chip was stored at 37˚C in a box with the chip covered in tape. Prior to droplet generation, the device was primed with fluorinated oil (3M, USA, Novec 7500) and connected to two programmable syringe pumps (Harvard Apparatus 11 Elite Series) located in a temperature-controlled room at 37˚C. One pump was fitted with a 1 ml syringe (BD, NJ, USA, #309628) having a 27-gauge needle (BD #305109), which was filled with 1.5% Pico-Surf 1 (Dolomite Microfluidics, MA, USA, 5% in Novec 7500) in Novec 7500 oil. In parallel, bacteria were diluted 1:2000 in 1% ultra-low-melt agarose containing Hystem-C hydrogel, yielding approximately $5 \times 10^6$ cells/ml, and the mix was loaded into a 1 ml syringe that was fitted onto a second pump. Polyethylene (PE/2) tubing (SCI COM, AZ, USA, #BB31695-PE/2) was attached to each syringe and then inserted into the appropriate ports on the microfluidics device (*Figure 1*). The pump holding the oil-phase was set to 700 µl/hour while the other pump was set to 400 µl/hour and droplets were collected in a 1.5 ml Eppendorf tube through tubing that had been inserted into the droplet collection port.

### Oil removal

100 µl of droplets was introduced into a 1.5 ml Eppendorf tube and 500 µl of 10% 1H,1H,2H,2H-Perfluoro-1-octanol (PFO) (Sigma #370533–25G) in Novec 7500 was added to the droplets. After mixing vigorously, the tube was placed in a centrifuge for 30 s at 250 RCF and 400 µl of 25% Extralink (ESI BIO #GS3006) in PBS was added to crosslink the hydrogel within the agarose. After crosslinking, the droplets were flicked into suspension and centrifuged for 30 s at 250 RCF. The PBS/droplet layer was transferred to a new 1.5 ml Eppendorf tube, subjected to centrifugation for 30 s at 250 RCF, and washed with PBS. The remaining droplets were resuspended in 2xYT broth.

## Droplet dispersal

Droplets were washed once in PBS at the indicated timepoints, incubated for 3 min at 50°C, and a mixture of agarase (Sigma #A6306), collagenase and hyaluronidase (StemCell Technology, MA, USA, #07912) was added to the samples for 45 min at 37°C. Samples were then incubated for 3 min at 60°C, subjected to centrifugation in a microfuge at 14 K RPM for 1 min. The supernatant was then aspirated, and the bacterial pellets were washed in PBS and prepared for flow analysis.

## Nitric oxide experiments

To analyze the response of colonies to exogenous NO, droplets containing *Y. pseudotuberculosis* were generated, oil was removed, and the droplets were rotated for 7 hr at 26°C in 2xYT broth to allow colony formation. After the 7 hr growth period, droplets were subjected to centrifugation for 30 s at 250 RCF and washed twice in PBS. 50 µl of droplets were transferred to a 1.5 ml Eppendorf tube and resuspended in 1 ml 2xYT broth and exposed to DETA-NONOate (Sigma #AC32865) during growth at 26°C with aeration for the indicated timepoints. For experiments performed at 37°C with 5% $CO_2$ (*Figure 5*), 50 µl of droplets were resuspended in 1 ml RPMI 1640 (Gibco, USA) supplemented with 10% FBS and 2 mM glutamine medium in 12- well non-tissue culture treated plates for the indicated times.

## Isolation of bone marrow-derived macrophages

Bone marrow cells were isolated from femurs, tibias, and humeri of female 6–8 week-old C57BL/6 mice, and terminally differentiated into macrophages in medium containing mouse CSF generated as previously described (*Auerbuch et al., 2009*). Bone marrow-derived macrophages (BMDMs) were subsequently frozen in fetal bovine serum (FBS) with 10% dimethyl sulfoxide (DMSO) and plated one day prior to incubation with droplets in 90% fresh medium (RPMI 1640 (Gibco) supplemented with 10% FBS and 2 mM glutamine) spiked with 10% sterile medium containing macrophage colony stimulating factor (MCSF) in 5% $CO_2$ at 37°C. Medium containing MCSF was conditioned by growth in the presence of mouse 3T3 fibroblasts that overproduce mouse MCSF at 37°C with 5% CO2, as previously described (*Auerbuch et al., 2009*; *Leber et al., 2008*). To stimulate BMDMs to induce iNOS expression, cells from frozen stocks were plated on non-tissue culture treated plates and allowed to recover overnight. The following day, BMDMs were stimulated with 200 U/ml IFNγ (Peprotech, NJ, USA) and 100 ng/ml LPS (Sigma) for 12–15 hr (*Mosser and Gonçalves, 2015*). For inhibition of iNOS, BMDMs were incubated with 2 mM NG-Monomethyl-L-arginine monoacetate (L-NMMA) (Acros Organics, NJ, USA) one day prior to priming with LPS and IFNγ (*Frawley et al., 2018*).

## Droplet-BMDM experiments

Droplets containing *Y. pseudotuberculosis* were generated, oil was removed, and the droplet-encased bacteria were grown with aeration for 7 hr at 26°C in 2xYT broth. After the 7 hr growth period, droplets were pelleted for 30 s at 250 RCF, washed twice in PBS and supernatant was removed. 50 µl of droplets were transferred to a well in a 12-well coverslip-bottom plate (Cellvis, CA, USA, #P12-1.5H-N), approximately 2 x $10^6$ BMDMs in RPMI 1640 supplemented with 10% FBS and 2 mM glutamine were added to the well, and the incubation proceeded at 37°C in the presence of 5% $CO_2$ for 4 hr. Samples were fixed in 2% paraformaldehyde (PFA) in PBS, washed in PBS, and fluorescent reporters were visualized by microscopic observation.

## Murine model of systemic infection

Six to 8 week old female C57BL/6 mice obtained from Jackson Laboratories (Bar Harbor, ME) were injected intravenously with $10^3$ bacteria. Three days post inoculation, spleens were removed and processed as described previously (*Crimmins et al., 2012*; *Davis et al., 2015*). All animal studies were approved by the Institutional Animal Care and Use Committee of Tufts University.

## Flow cytometry

After microcolony dispersal, samples were fixed in 2% PFA in PBS, washed in PBS, and filtered through 40 µm filters. Fluorescent reporters were detected using a Bio-Rad S3e cell sorter. 100,000 events were collected for each sample.

## Fluorescence microscopy

After 4 hr of nitric oxide exposure or challenge with BMDMs, droplets were fixed in 2% PFA in PBS for 10 min. Droplets that were exposed to the NO donor were mounted in eight well chamber slides using ProLong Gold (Life Technologies, CA, USA). Droplets that were incubated with BMDMs were visualized in 12-well coverslip-bottom plates (Cellvis). For visualization of BMDMs, cells were fixed in 2% PFA in PBS, permeabilized for 30 s in ice cold methanol and non-specific binding was blocked using 2% BSA in PBS. Cells were stained with rabbit anti-mouse iNOS (Abcam, MA, USA) at a dilution of 1:100 and goat anti-rabbit-Alexa 488 (Invitrogen, CA, USA) at a dilution of 1:500. Nuclei were stained with Hoechst at a dilution of 1:10,000. To analyze tissue samples, C57BL/6 mice were inoculated intravenously with the WT *gfp+ Phmp::mcherry* strain. Three days post inoculation, spleens were harvested and fixed in 4% PFA in PBS for 3 hr, tissue was frozen embedded in Sub Xero freezing medium (Mercedes Medical, FL, USA) and cut into 10 μm sections using a cryostat microtome (Microm HM505E). To visualize reporters, sections were thawed in PBS, stained with Hoechst at a 1:10,000 dilution, washed in PBS, and coverslips were mounted using ProLong Gold (Life Technologies). Tissue was imaged with a 63x objective on a Zeiss Axio Observer.Z1 (Zeiss) microscope with Colibri.2 LED light source, an Apotome.2 (Zeiss) for optical sectioning, and an ORCA-R$^2$ digital CCD camera (Hamamatsu) (*Crimmins et al., 2012*; *Davis et al., 2015*).

## Determination of droplet size, bacterial counts and microcolony area

For determination of droplet size, images were captured by phase contrast microscopy using a 20x lens, and images were analyzed to identify 300 droplets, with analysis in Volocity. Droplet size was determined by identifying a threshold that defines edges of droplets and determining the diameter in pixels. The data were then converted to metric scale using a stage micrometer and displayed as a histogram. To determine bacterial counts, images were captured by phase contrast and fluorescence microscopy using a 20x lens and the number of bacteria/droplet was manually counted in 500 droplets. To determine microcolony area, images were captured by phase contrast and fluorescence microscopy using a 20x lens with analysis in Volocity. Microcolony area was determined by identifying a threshold that defines edges of microcolonies in the GFP channel and determining the number of pixels in the region of interest (ROI). The data were then converted to metric scale using a stage micrometer and displayed as μm$^2$.

## Quantitative mage analysis

MATLAB scripts were written to calculate the average intensity of fluorescence about the periphery of microcolonies and point of lowest fluorescence (https://github.com/isberg-lab/droplet; *Shull et al., 2019*; copy archived at https://github.com/elifesciences-publications/droplet). Periphery to point of lowest intensity (PLI) ratios were calculated by generating a mask representing the periphery of a defined width of the microcolony, and a mask representing the PLI of a defined radius. The periphery mask was generated by first thresholding the green channel using Otsu's method and converting the resulting matrix into a binary matrix. The binary matrix was morphologically opened with a disk-shaped structuring element of radius two and then morphologically closed with a disk-shaped structuring element of radius four. Morphologic opening and closing eliminate dim artifacts and fuse the microcolony mask into a single region of interest (ROI). Any remaining holes within contiguous ROIs are filled. ROIs touching the edge of the field are removed, as are all but the largest ROI in the field. This ROI, covering the entire microcolony, is eroded with a disk-shaped structuring element of radius 12 and complemented. The initial mask and complemented eroded mask are combined to achieve a binary mask of the periphery of the object. The number of pixels and the total intensity in the resulting mask was recorded. The mask was applied to the raw image data in the red channel, and the total intensity was recorded. Total intensity of the red was normalized to the number of pixels to achieve an average intensity for the entire periphery. The average intensity of the red channel was normalized to the average intensity of the green channel. To calculate the PLI, the determined eroded mask was applied to the red channel to isolate the pixels belonging to the microcolony while excluding those that are part of the periphery. A Gaussian blur of sd = 2.0 was applied to reduce the effects of low-intensity noise introduced by the imaging process. The coordinates of the minimum value of this image were calculated, generating a binary matrix with value = 1 at the coordinates of the PLI and zero at all other points. This matrix was

morphologically dilated with a structuring element of radius five to sample a region of lowest intensity rather than a single pixel. The number of pixels in this resulting mask are then recorded and the total intensity of the green channel was recorded. The mask was applied to the raw image data in the red channel, and the total intensity recorded. Total intensity was normalized to number of pixels to achieve an average intensity for the entire PLI region in the red and this value was normalized to the average intensity of the green channel.

## Acknowledgements

The work in this study was supported by NIAID grants R01 AI110684 and R21 151593 to RRI and U01AI124302 and R01 AI110724 to TvO. SC and LS were supported by predoctoral training grant T32 TM007310 from NIGMS. We thank the Boston College Integrated Sciences Nanofabrication Facility for support and thank Kristen Davis, Efrat Hamami and Wenwen Huo for review of the manuscript.

## Additional information

### Funding

| Funder | Grant reference number | Author |
|---|---|---|
| National Institute of Allergy and Infectious Diseases | R01 AI110684 | Ralph Isberg |
| National Institute of Allergy and Infectious Diseases | R21 151593 | Ralph Isberg |
| National Institute of Allergy and Infectious Diseases | U01 AI124302 | Tim van Opijnen |
| National Institute of Allergy and Infectious Diseases | R01 AI110724 | Tim van Opijnen |
| National Institute of General Medical Sciences | T32 TM007310 | Stacie A Clark Derek Thibault |

The funders had no role in study design, data collection and interpretation, or the decision to submit the work for publication.

### Author contributions

Stacie A Clark, Conceptualization, Data curation, Formal analysis, Validation, Investigation, Visualization, Methodology, Writing - original draft, Writing - review and editing; Derek Thibault, Conceptualization, Methodology; Lauren M Shull, wrote computer code scripts for the experiments; Kimberly M Davis, Conceptualization, Formal analysis, Writing - review and editing; Emily Aunins, Investigation, Methodology; Tim van Opijnen, Conceptualization, Formal analysis, Methodology, Writing - review and editing; Ralph Isberg, Conceptualization, Formal analysis, Funding acquisition, Investigation, Methodology, Project administration, Writing - review and editing

### Author ORCIDs

Stacie A Clark (iD) https://orcid.org/0000-0002-9638-199X
Tim van Opijnen (iD) http://orcid.org/0000-0001-6895-6795
Ralph Isberg (iD) https://orcid.org/0000-0002-8330-3554

### Ethics

Animal experimentation: This study was performed in strict accordance with the recommendations in the Guide for the Care and Use of Laboratory Animals of the National Institutes of Health. All of the animals were handled according to approved institutional animal care and use committee (IACUC) protocols (#B2016-21 and B2019-03 ) of Tufts University. No surgeries were performed.

Decision letter and Author response
Decision letter https://doi.org/10.7554/eLife.58106.sa1
Author response https://doi.org/10.7554/eLife.58106.sa2

## Additional files

### Supplementary files
• Supplementary file 1. P-values_*Figure 6–8*.

• Transparent reporting form

### Data availability
All data generated or analysed during this study are included in the manuscript and supporting files. Scripts described have been deposited with GitHub and the appropriate link is provided in manuscript.

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
