## [Decision Letter]

**Acceptance summary:**

In the host, bacteria often form clonal microcolonies that display heterogeneous behavior in response to their spatial organization with respect to host immune attack. This manuscript reports on the creation of a microdroplet-based microfluidics model allowing for the capture of bacteria-bacteria as well as host-bacteria interactions occurring usually in the deep tissues of the infected host. The work provides the community with an interesting alternative in between pure tissue culture and animal models which are in themselves often limiting in being either too simple for the former and too complex for the latter. Importantly, many features of bacterial cells are accessible in the model reported herein and this microdroplet-based microfluidics model should prove very useful for many research groups.

**Decision letter after peer review:**

Thank you for submitting your article "Topologically correct synthetic reconstruction of pathogen social behavior found during*Yersinia*growth in deep tissue sites" for consideration by *eLife*. Your article has been reviewed by three peer reviewers, one of whom is a member of our Board of Reviewing Editors, and the evaluation has been overseen by a Senior Editor. The following individual involved in review of your submission has agreed to reveal their identity: Neeraj Dhar (Reviewer #2).

The reviewers only have a few minor suggestions.

Reviewer #1:

In this report, Clark et al. present the creation of a microdroplet-based microfluidics model recapitulating the bacteria-bacteria as well as host-bacteria interactions occurring usually in deep tissues of the infected host. In the host, bacteria often form clonal microcolonies that display heterogeneous behaviour in response to spatial organisation with respect to host immune attacks. Authors show that in such a model, Yersinia pseudotuberculosis can grow in microcolonies, which, similarly to what the group had observed in other experimental set ups, rely on expression of Hmp mostly by peripheral bacterial cells for resilience of the entire population to added NO. Importantly, such model is also shown to be amenable to addition of bone marrow derived macrophages to the droplets, mimicking thus the situation in tissues.

This is a great report, easy to read and well presented. The model has been extensively tested by the authors who provide the community here with an interesting alternative in between pure tissue culture models and animal models which are in themselves often limiting in being either too simple for the former and too complex for the latter. Importantly, many features of the bacterial cells are accessible in such model and it is going to be very useful to many research groups.

I have no essential revisions to request, the manuscript is an excellent piece of work.

Reviewer #2:

In this study, the authors have attempted to build a system that allows them to capture microbial behavior and its interactions with host immune cells. Yersinia bacteria typically replicates in intestinal lumen and lymph nodes forming microcolonies surrounded by neutrophils, macrophages and other immune cells. Previous in vivo studies have shown that there is a topological induction of detoxifying proteins (*hmp*) in the bacterial microcolonies with the bacteria in the periphery of the microcolony showing the highest induction. To better study these processes and with the aim of developing a system amenable for development of high-throughput screens, the authors employ droplet-based microfluidics to facilitate the growth of bacteria in gel-based matrix. Initially they characterize these droplets and demonstrate exponential growth of the bacteria in these droplets, where they form microcolonies. These microcolonies were then exposed to different concentrations of NO and the important role of *hmp* in resisting this stress was confirmed. Using a fluorescent reporter for *hmp* expression, the heterogeneous and spatially organized expression is demonstrated in wild-type strains using FACS and microscopy, which is lost in strains deleted for *hmp*. Similar pattern of expression is seen when the droplets containing bacteria are coated with macrophages activated to express NOS and is similar to the expression seen in sections of spleens from infected mice. This peripheral induction of *hmp* is lost when the NOS response is inhibited using chemical inhibitors of NO.

While there are no new observations on the pathogenesis or host-microbe interactions aspect, this study presents the development of a novel in vitro system that facilitates future studies on spatial relationships between immune cells and bacteria in a more amenable manner. It might be a good fit for the Tools section. Overall this is a nice and elegant study and the manuscript and data has been presented well.

I do not have any major issues with the manuscript. I do have a few suggestions that might improve the presentation.

– If I understood correctly, the reporter carries the fluorescent marker on a episomal plasmid. Variation in copy number potentially could influence their readouts for heterogeneity in gene induction. The authors could comment on this briefly in the Discussion.

– Figure 5 could be moved to the supplementary data.

Reviewer #3:

In this study by Isberg and colleagues, the authors provide to the community a very interesting experimental platform. The study appears motivated by the years-long interest of the Isberg lab in the pathogenesis of Yersinia. These bacteria create microcolonies in several tissues of an infected animal, the significance of which has only been explored in theory. The major question related to these microcolonies is that of the social behavior of individual bacteria.

Herein, the authors present a microfluidics system to create droplet-encased bacteria, which form clusters that resemble those identified in vivo. The major advance provided here is the demonstration that these synthetic clusters recapitulate behaviors observed in vivo, as they relate to NO responsiveness. The source of the NO could be chemical or macrophage derived, and the NO induces the expression of NO-responsive genes in bacteria that are present in the periphery of the cluster.

The potential for using this platform to ask "sufficiency questions" related to immune signals that influence intra-cluster bacterial behavior is powerful and should be embraced by the community.

While there are several experimental questions that I would like answered, they all would require an extension of this work that is already of sufficient quality to attract attention from the community.

---

## [Author Response]

Reviewer #2:[…] While there are no new observations on the pathogenesis or host-microbe interactions aspect, this study presents the development of a novel in vitro system that facilitates future studies on spatial relationships between immune cells and bacteria in a more amenable manner. It might be a good fit for the Tools section. Overall this is a nice and elegant study and the manuscript and data has been presented well.I do not have any major issues with the manuscript. I do have a few suggestions that might improve the presentation.– If I understood correctly, the reporter carries the fluorescent marker on a episomal plasmid. Variation in copy number potentially could influence their readouts for heterogeneity in gene induction. The authors could comment on this briefly in the Discussion.

We have added text to the fourth paragraph of the Discussion in this regard. Noted in this section is the fact that we used a chromosomally-integrated reporter in a previous publication and it behaved identically to the plasmid-borne reporter.

– Figure 5 could be moved to the supplementary data.

We decided to keep Figure 5 in place, since it is an important strategy that should be used in future image analysis and is central to understanding how we quantitate our data.